# DisRnet: Discrete Relevance semantic segmentation network based on Discrete Data Distribution

## ABSTRACT

In the current pixel-level semantic segmentation tasks, the current popular algorithm models are generally based on CNNs and combine contextual information to achieve semantic segmentation of images. However, when these algorithms extract features, they are all affected by the mean pooling layer or the maximum pooling layer, which causes the extracted features lose some spatial information easily. Aiming at these problems, this paper designs a discrete pooling layer (Dis pool) and a correlation pooling layer (Rel pool) by using the characteristics of discrete data distribution. The Dis pool can retain the spatial location information of features through the discrete characteristics of discrete data. The Rel pool can utilize the correlational information between the discrete data to preserve the correlation between the features. Then, DisRnet is designed by fusing the Dis pool and Rel pool on the residual structure. Finally, under the Cityscapes, SBD datasets and Pascal VOC dasets, compared with some SOTA models, it is verified that DisRnet has superior performance.

**Keywords**: Discrete Data Analysis, Residual structure, Pool layer, Image features, Related Features, Discrete features.

**Index Terms:** Human-centerd computing- Visualization – Visualization empirical studies, theory

## 1 INTRODUCTION

Semantic segmentation is a hot topic in computer vision. Likewise, pixel-level semantic segmentation is an important and complex task, and feature extraction in it is also a difficult task. In the traditional feature extraction algorithm, the Roberts[1] and Prewitt operators extract the edge feature information of the image through the first-order difference, and the Sobel operator obtains the edge feature information of the image through the second-order difference. In image segmentation, Sobel operator is superior to Roberts and Prewitt operator. On the basis of the Sobel operator, the Robinson operator introduces convolution kernels in 8 directions to ensure that the extracted information is more accurate. However, the parameters in traditional algorithms are fixed, so the generalization ability of these algorithms is relatively weak.

On the contrary, CNNs[2-5] algorithms have achieved excellent results in image classification, segmentation, tracking of Kaggle[8] competition or AI Challenger competition. FCN[9] uses deconvolution for up-sampling to make the extracted features more detailed. U-Net[10-11] uses the network symmetric structure to fuse high-dimensional features and low-dimensional features to strengthen edge features and make the segmentation effect more superior. In CPFNet[12], the dilated convolution[54] can expand the field of the convolutional layer to extract more feature information, and combine the inception module[4] to achieve context feature fusion. Finally, CPFNet achieve superior results in medical datasets. STDC[13] integrates multiple scales on the basis of FPN[14-15], and its performance is superior to the CPFnet algorithm. BiseNetV2[16] adopts a bilateral segmentation structure on the basis of STDC, namely Detail Branch and Semantic Branch. Detail Branch obtains more low-level feature information by expanding the channel, and Semantic Branch expands the receptive field to obtain high-level feature information through a lightweight convolutional layer. BiseNetV2 can solve the problem of structural redundancy. Although the CNN-based algorithm has high accuracy, it is common that the extracted features will lose a lot of spatial information due to the pooling layer, which eventually happen structural redundancy, large computational load, and segmentation errors in the semantic segmentation network.

Here we design Dis pool and Rel pool by exploiting the distribution properties of discrete data. The Dis pool extracts the discrete coefficient and the spatial position of the coefficient by analyzing the data distribution characteristics of the discrete data. Therefore, the Dis pool can retains the spatial position information and discreteness of image features. Similarly, the Rel pool extracts the correlation coefficient of the feature by analyzing the correlation between the data and the data, thereby retaining the data correlation of the feature. Compared with traditional pooling layers, Dis pool and Rel pool can effectively preserve the spatial information of features. Here we design the DisRnet model by introducing the residual structure[23] based on the Dis pool and Rel pool. Among them, the residual structure mainly realizes the fusion of context information, which makes the model more lightweight on the one hand, and improves the accuracy on the other hand. Under the datasets of Cityscapes[17], SBD[18], and PASCAL VOC[26], compared with the SOTA algorithm, DisRnet has certain advantages in accuracy and speed.

Our main contributions are highlighted as follows:
1. In this paper, Dis pool and Rel pool are designed using statistical analysis methods for discrete data.
2. Based on the Dis pool and Rel pool, the residual structure is used to achieve context information fusion, which not only makes the model more lightweight, but also effectively improves the accuracy.

3. DisRnet is compared with the SOTA algorithm under three datasets. The effectiveness of DisRnet is comprehensively demonstrated from the four indicators.

## 2 RELATED WORK

As we all know, an image is composed of an indeterminate number of pixel values according to certain rules, and the values is from 0 to 255. Although the rules are complex, the image data is discrete, so when we perform image processing, we can obtain some features by means of statistical analysis of discrete data. Traditional algorithms can use some functions to analyze the statistical characteristics of discrete data and extract simple features, such as SIFT[19], SURF[20], HOG[21], DOG, LBP[22], Haar, etc. However, the functions of these algorithms will use fixed parameters, which resulting in these algorithms having certain limitations in image processing and cannot be popularized. That is, these algorithms can only process some images with very obvious local features or images in a specific scene.

In the CNNs algorithm, the convolution layer uses many variable parameters to perform simple inner product calculation on the image, and uses adaptive adjustment to adjust and optimize those parameters. Therefore, with the help of multiple parameters, the generalization ability of the convolutional layer is stronger than that of the traditional algorithm. The pooling layer extracts the maximum pixel value or the average pixel value of feature from sliding window. Although the pooling layer has the function of dimensionality reduction, a lot of information will be lost in the process of dimensionality reduction. The STL[26] analyzes the distribution of low-level texture features, and achieves an effective semantic segmentation effect. Then, the paper use the context information fusion method to achieve the segmentation effect on the extracted features.

Inspired by STL[26], we extract discrete features and related features as important features by analyzing the statistical properties of discrete data. Then we fuse these extracted features on the structure which designed based on ResNet[25] to achieve contextual information fusion. Finally we design the DisRnet network. In terms of model size, DisRnet utilizes the network characteristics of ResNet to realize the lightweight of the model. In feature extraction, DisRnet uses Dis pool and Rel pool to preserve spatial information without loss, thereby improving the accuracy of the model.

## 3 METHOD

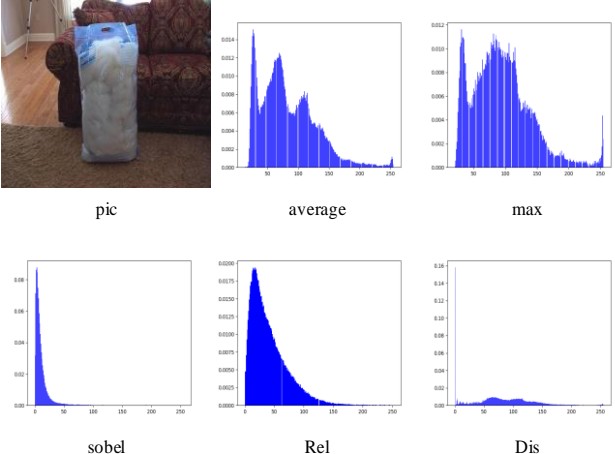

Figure 1: pic is a picture in the SBD dataset. average, max, sobel, rel, and dis are the histograms processed by their respective algorithms.

Here we design a Dis pool and a Rel pool inspired by STER[24] and STL[26]. From the histogram in Figure 1, we can see that compared with the results of the max pooling layer and the mean pooling layer, the features extracted by the Dis pool and Rel pool are smoother and more uniformly distributed. Therefore, the features extracted by the Dis pool and Rel pool are also easy to calculate.

### 3.1 Dis pool

The image $P$ with dimension H*W gets n feature maps under the sliding window, namely $\{P_1, P_2, P_3,...P_n\}$. The dimension of these feature maps is h*w, and is same of the size of the sliding window, as show in Figure2. $P_m$ obtains the corresponding discrete value $d_m$ and mean $a_m$ under formulas 1 and 2.

$$a_m = \sum_{i,j} P_{i,j} \tag{1}$$

$$d_m = \sum_{i,j} P_{i,j} - a_m / i * j \tag{2}$$

Both $d_m$ and $a_m$ describe the data characteristics of $P_m$. The pixel value of $P_m$ is processed under formula 3, and the spatial position of discrete features can also be located, where $\lambda$ is a hyper parameter. $P^{i,j}_m$ is the value of $P_m$ at (i,j) position. If $P^{i,j}_m$ is less than $\lambda*a_m+d_m$, it means that the value is lower than the discrete value range, and then the value of this position is changed to 0, otherwise it is 1, finally a new feature map $C_m$ is generated. All feature maps $C_m$ in the feature group $\{C_1, C_2, C_3,...C_n\}$ that exist at the position (i, j) will get the values $l_{i,j}$ of the position under formula 4.Then a new feature map $l$ is also generated, where $\lceil * \rceil$ means to take the largest integer less than *.

$$c_m^{i,j} = \begin{cases} 0, & C_m^{i,j} < \lambda * a_m + d_m \\ 1, & C_m^{i,j} \geq \lambda * a_m + d_m \end{cases} \tag{3}$$

$$l_{i,j} = \sum_m c_m^{i,j} / (\lceil \frac{h}{s} \rceil * \lceil \frac{w}{s} \rceil) \tag{4}$$

$$p = l * P \tag{5}$$

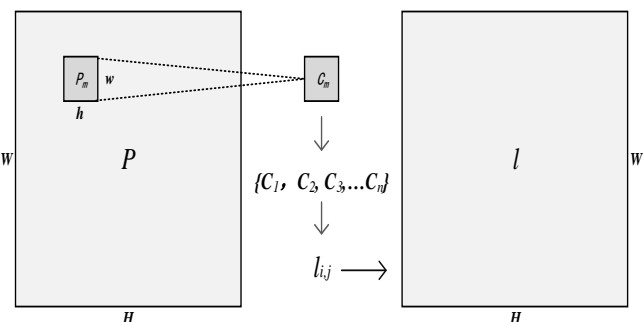

Figure 2: Schematic of the Dis pool. H, W represent the size of the input picture $P$. h, w is the size of the sliding window. $P_m$ is the pixel data read in $P$ by the sliding window.

The input image $P$ gets the feature map $l$ through Dis pool. The feature map $l$ retains the spatial position of the discrete features of the image. In order not to lose pixel information, the image $P$ is then multiplied by l, as shown in formula 5. Therefore, the Dis pool has the ability to retain the spatial information of the image effectively, which can reduce the value of feature map, reduce the computational effort, and improve the computational efficiency.

## 3.2 REL POOL

The Dis pool can retain the spatial location information of the feature values in the above-mentioned manner. Here, in order to maintain the correlation invariance between feature values, the Rel pool is proposed here. From the theory of covariance and Pierce correlation coefficient[55], we know that there is no correlation and covariance between two independent discrete data, so we stipulate that the step size s of the sliding window should be smaller than the size of the window, that is (s <h,s<w).

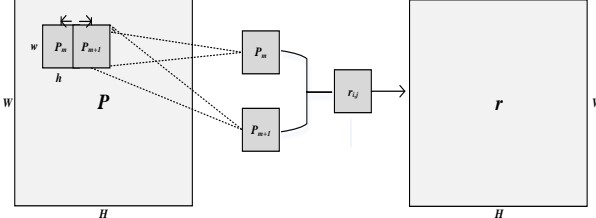

Figure 3: Schematic of the Rel pool. H, W represent the size of the input picture *P*. h, w is the size of the sliding window. $P_m$ is the pixel data read in *P* by the sliding window. s is the step size of the sliding window

The input image P with dimension H*W gets n feature maps under the sliding window, namely $\{P_1, P_2, P_3,...P_n\}$. Through formula 6, we can obtain the correlation coefficient $r_{i,j}$ between the feature maps $P_m$ and $P_{m+1}$, which expresses the correlation between the two feature maps, and the larger the correlation coefficient, the greater the correlation. Finally, the feature values $r_{i,j}$ of the n positions form a new feature map *r*, which ensures the

correlation between the feature values.

$$r_{(i,j)} = \frac{\sum(P_m|(i,j) - \overline{P}_m|(i,j))(P - \overline{P}_{m+1})}{\sqrt{(\sum(P_m|(i,j) - \overline{P}_m|(i,j))^2)(\sum(P_{m+1} - \overline{P}_{m+1})^2)}} \quad (6)$$

$$n = \left\lceil \frac{H - h + 2*p}{s} + 1 \right\rceil * \left\lceil \frac{W - w + 2*p}{s} + 1 \right\rceil \quad (7)$$

$$R = r * P \quad (8)$$

where $P_m|(i,j)$ represents the feature map with (i,j) as the upper left point, and $\overline{P}_m|(i,j)$ represents the mean of the feature map Pm. $P_{m+1}$ represents the next feature map of the $P_m$ feature map. The feature map *r* is dot-multiplied with the data *P*, and finally the feature map *R* pooled by the Rel pool is obtained.

## 3.3 DisRnet

Rel pool can obtain the correlation between the feature values in the feature map and ensure the correlation invariance between the feature values. At the same time, the feature map obtained by Dis pool can preserve the discreteness of the feature values and the spatial invariance of some the feature values, thereby preserving the stability of edge features. Here we fuse the Rel pool and the Dis pool on the basis of the residual structure and introduce the upsampling[27-29] layer to design DisRnet. The DisRnet structure is shown in the Figure 4 below. The parameters of the DisRnet network structure are shown in Table 4 below.

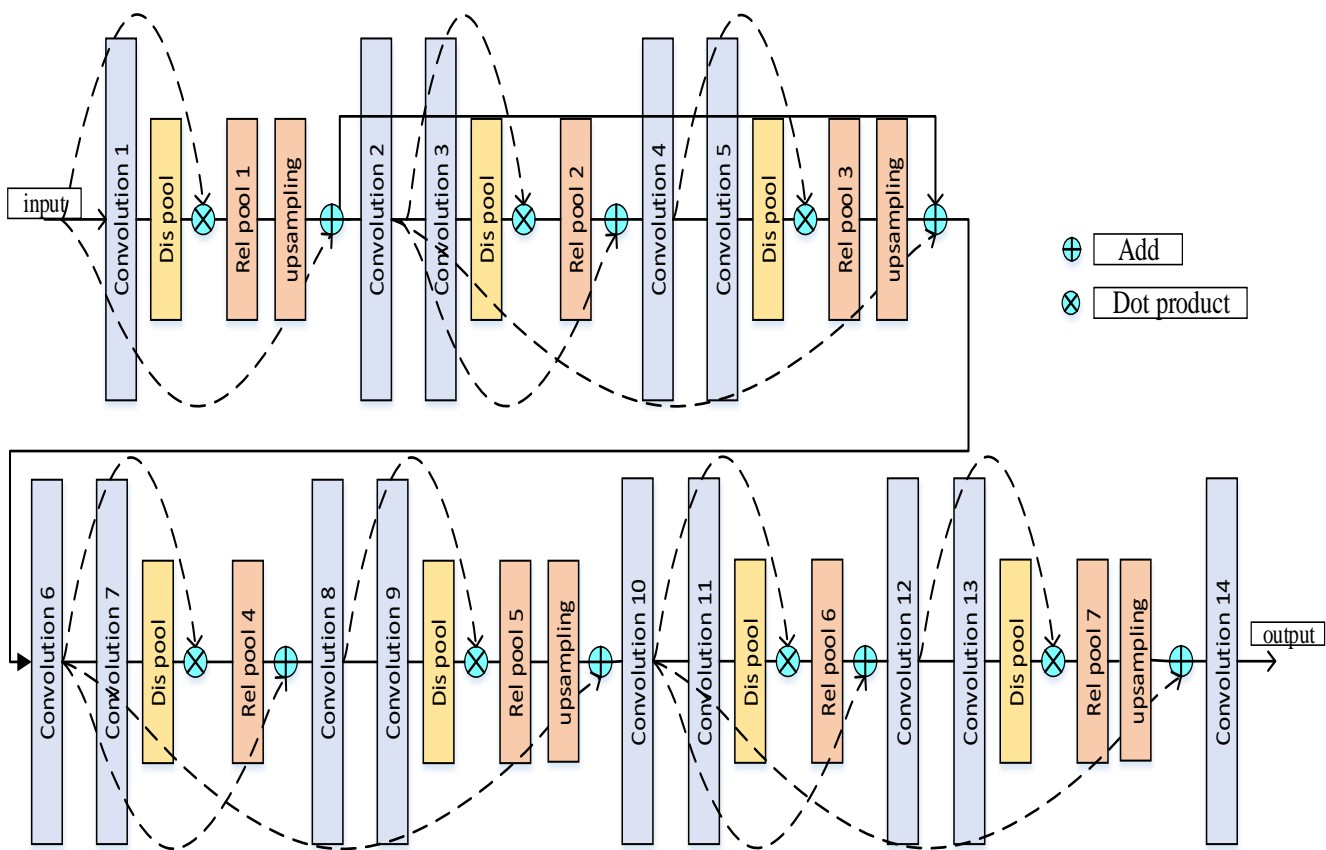

Figure 4: DisRnet's networking framework.

Table 1. The network parameters of.DisRnet. Conv3/5/7/9/11/13 represent Convolution3, Convolution5, Convolution7, Convolution9, Convolution11, and Convolution3 in Figure 4, respectively.

| Layer | Kernel/stride | Parameters |
|-------|---------------|------------|
| Conv1 | 128*11*11/4 | 15488 |
| Dis pool | 3*3/1 | - |
| Conv3/5/7/9/11/13 | 8*1*3/2
8*3*1/2 | 3072 |
| Rel pool | 3*3/2 | - |
| Conv4/8/12 | 16*3*3/2 | 1152 |
| Conv2/6/10 | 64*11*11/4 | 61952 |
| Conv14 | 1*1*1/1 | 8 |

The input data is multiplied by the feature map *l* obtained by Dis pool, which the purpose is to locate the spatial position of the edge feature through the discrete coefficient feature map *l*. Then, the result is passed to Rel pool, and finally the obtained result is added to the input data, which can strengthen the information of edge features on the one hand, and preserve the correlation between features on the other hand. Therefore, the dot product is to calibrate the spatial position of the edge features. The purpose of addition is to strengthen the information of the edge features and preserve the correlation between the features. The Resnet residual structure is referenced in the DisRnet structure, which can effectively extract edge features on the one hand, and make the model more lightweight on the other hand.

## 4 EXPERIMENT

### 4.1 DATASETS AND ENVIRONMENT

Cityscapes contains a total of 5000 fine images, of which 2975 are training images, 500 validation images and 1525 testing images. An additional 20k images with Coarse's rough annotations are included. And the performance of the algorithm is evaluated on the average precision metric of the 8 semantic classes of the dataset.

The Pacal VOC (07+12) dataset is divided into 4 categories and 20 sub-categories. There are 33k photos in total, which including 16k train+val datasets and 16k test datasets.

SBD belongs to the augment dataset of the VOC dataset. It contains 11355 labeled images in VOC. We still use the segmentation of the VOC dataset as an annotation to train the network. SBD divides the total dataset into two parts, 8498 training images and 2857 testing images.

We are experimenting based on the version of Keras-1.2+TensorFlow 0.12.1. The corresponding hardware platform is GTX1070+CPU AMD Ryzen7.

For network training, we choose ADAM[32] as the optimizer with a momentum of 0.65 and a weight decay of $2.5e-4$ during training. The initial learning rate is set to 0.8. We choose different batch sizes for the three datasets, 6 for Cityscapes, 12 for Pascal VOC, and 8 for SBD. We set the maximum training epoch to 1000. To increase the diversity of training, we apply data augmentation methods. It contains random horizontal flips, mean subtraction on the input image. In addition, DisRnet uses a 1*1 convolution format in the last layer, and we use the following function to calculate the loss function.

$$\bar{a} = \sum_x a \qquad (9)$$

$$b = a - \lambda * \bar{a} + 1 * e^{-4} \qquad (10)$$

$$z = -\frac{1}{n} * \sum_x [d \ln b + (1-d)\ln(1-b)] \qquad (11)$$

In the above formula, x represents the sample, d represents the actual label, a represents the predicted output, and n represents the total number of samples. In the forward propagation of DisRnet, the input data a will be numerically determined by the parameter $\lambda$ after passing through the Dis pool. Here, in order to reduce the difference between the predicted data and the real data, in formula 10, let the input data a subtract the mean value of $\lambda$ times, so that the predicted value is smoother. And formula 11 is the cross entropy loss function[33].

### 4.2 ABLATION EXPERIMENT

We conduct ablation experiments under the SBD dataset to demonstrate the feasibility of Dis pool and Rel pool. First, we only extract the convolutional layer of ResNet-18 as the basic network skeleton, and combine the Dis pool, correlation pooling layer, max pooling layer and mean pooling layer respectively to generate a new network structure. These new network structures are each trained on the SBD training set for 20 epochs and then tested on the test set. The result is show in Figure 5 and Table 2.

Table 2. Results of ablation experiments. ResNet-18 represents the network structure after removing the pooling layer

| Model | mIou(%) |
|-------|---------|
| ResNet-18 | 40.6 |
| +Dis pool | 44.3 |
| +Rel pool | 42.7 |
| +Max pool | 41.0 |
| +Ave pool | 43.2 |

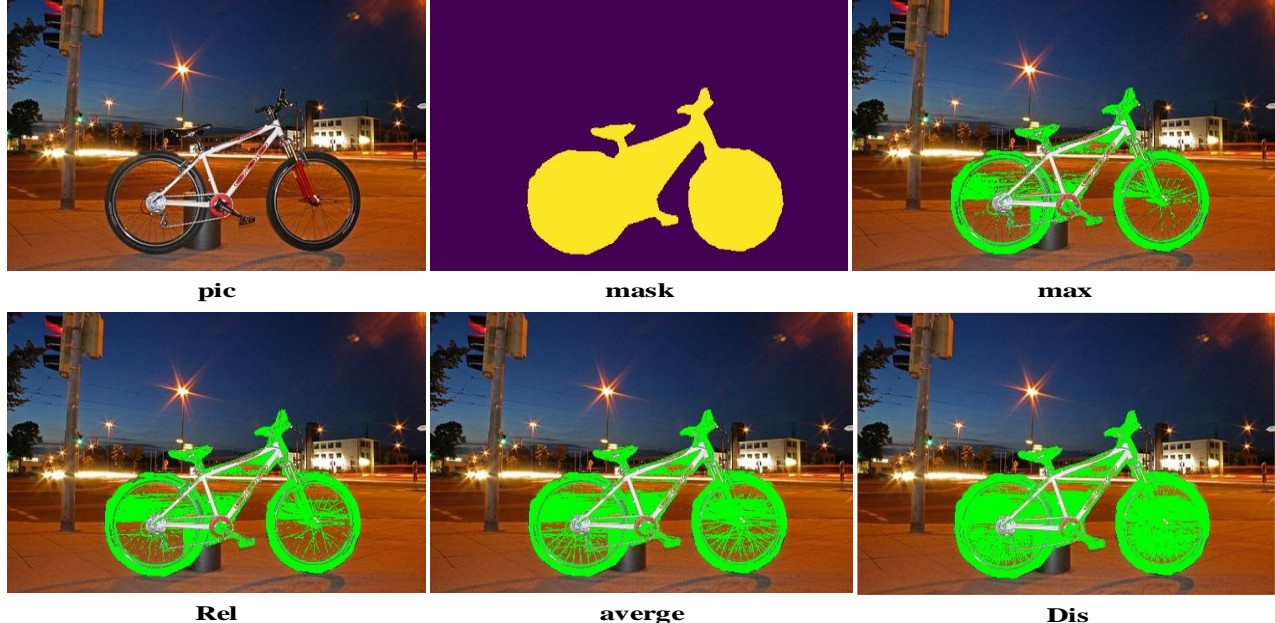

|  |  |  |
|:---:|:---:|:---:|
| pic | mask | max |
| Rel | averge | Dis |

Figure 5: pic is a picture from the SBD dataset, and mask is the segmentation map of the picture. max, Rel, average, Dis correspond to the segmentation maps of the respective pooling layers in Table 2.

It can be seen from the accuracy indicators in Table 2 that the results obtained by the Dis pool and the mean pooling layer have achieved the optimal value, but the Resnet18 without any pooling layer obtained the lowest value. Combined with the segmentation effect in Figure 5, we can intuitively see that the segmentation effect obtained by the Dis pool and the Rel pool is significantly better than the average layer and the max layer. Among them, when the feature extraction is performed on a target with a large local feature area, the Rel pool will cause some feature weight ratios to be too high due to the high correlation around the target, which will resulting in inaccurate feature segmentation and other problems. However, compared with the loss of pixel spatial information caused by the average layer and the max layer, the Rel pool can solve the spatial correlation information around the pixels well. For problems with different weight ratios of some features, the Rel pool can effectively weight the features through the convolution layer. The Dis pool can locate the specific position of the target edge feature through the discrete degree around the pixel and achieve the segmentation effect of the target. And then the feature from Dis pool will calibrates the correlation of the target feature through the Rel pool to complete the accurate segmentation of the target. From the accuracy comparison in Table 2 and the effect comparison in Figure 5, it can be concluded that the combination of Dis pool and Rel pool can achieve a good segmentation effect.

### 4.3 HYPER-PARAMETER

We use $\lambda$ in the Dis pool, the main purpose is to discretely determine the feature values, so the $\lambda$ can indirectly determine the accuracy of the entire network. Here we assign different values to $\lambda$ on DisRnet and compare them under the Cityscapes dataset, as shown in the following Table 3 and Figure 6.

Table 3. Accuracy for different values of $\lambda$

| $\lambda$ | 0.7 | 0.1 | 0.13 | 0.16 | 0.19 | 0.21 |
|---|---|---|---|---|---|---|
| MIoU(%) | 0.705 | 0.755 | 0.787 | 0.753 | 0.687 | 0.672 |

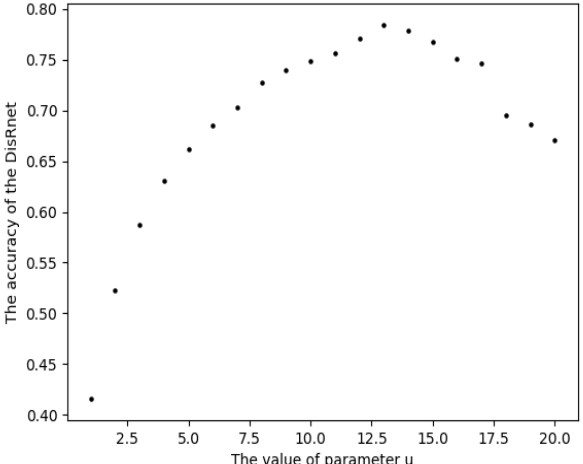

Figure 6: DisRnet's accuracy for different values of $\lambda$

From Figure 6, we can intuitively see that the accuracy of DisRnet varies with the value of a. But when $\lambda$ is equal to 0.13, the accuracy of DisRnt achieves the best value. Then we combine the data in Table 3 to determine that the parameter of $\lambda$=0.13 is used in this paper.

### 4.4 COMPARED WITH SOTA

In order to verify the efficiency of DisRnet, we compare DisRnet network with SETR, STL, CLL[35], GUDA[36], CFPnet, Deep[34], BiseV2, DFAnet[45], ESE[42-43], STS[44], PANet[46] in Cityscapes dataset, PASCAL VOC dataset and SBD dataset.

#### 4.4.1 CITYSCAPES

The test set of Cityscapes contains 6 categories, which in turn contain 33 sub-categories. From the perspective of sub-categories accuracy, DisRnet and SETR are optimal. Both DisRnet and

SETR use the context information fusion method, so the accuracy of DisRnet and SETR is higher in urban scenes. Compared with the features extracted by Bisev2 and SETR, DisRnet obtains discrete features and related features in the image by analyzing the statistical characteristics of discrete data, so that making the features more comprehensive and covering more information, so the accuracy rate is also better.

Table 4.  The results of DisRnet and 13 SOTA algorithms under the Cityscapes dataset are compared.

| Method | road | wall | bridge | pole | sky | person | car | bus | truck | terrain | bicyle | tunnel | mIou |
|--------|------|------|--------|------|-----|--------|-----|-----|-------|---------|--------|--------|------|
| DisRnet | 40.6 | 70.3 | **86.7** | 70.1 | 65.8 | 55.2 | 46.9 | **65.7** | 77.2 | **86.1** | **77.4** | **80.9** | **75.4** |
| SETR | 67.7 | 69.6 | 74.8 | 61.9 | **82.9** | **73.6** | **86.5** | 64.5 | **92.3** | 55.2 | 62.4 | 51.1 | 73.4 |
| GUDA | 60.2 | 65.1 | 73.6 | **86.6** | 44.0 | 57.5 | 48.1 | 63.6 | 83.7 | 69.1 | 70.8 | 45.9 | 68.6 |
| FedDG | 44.3 | 52.1 | 55.3 | 31.2 | 51.5 | 52.4 | 44.0 | 37.7 | 58.7 | 35.2 | 47.9 | 33.7 | 47.2 |
| DPL | 66.2 | 54.2 | 45.3 | 36.5 | 30.4 | 54.8 | 20.4 | 40.1 | 34.6 | 58.3 | 68.7 | 51.2 | 53.6 |
| CFPnet | 44.3 | 73.2 | 66.2 | 62.1 | 53.4 | 38.6 | 62.5 | 61.7 | 67.6 | 60.2 | 78.7 | 66.3 | 66.8 |
| Deep | 11.6 | 32.3 | 16.5 | 39.4 | 40.4 | 18.6 | 31.3 | 43.3 | 20.6 | 17.9 | 38.1 | 25.6 | 38.2 |
| Bisev2 | 47.4 | **79.1** | 53.5 | 65.9 | 81.0 | 71.5 | 65.1 | 35.2 | 72.6 | 63.8 | 73.5 | 65.6 | 71.5 |
| DFAnet | **69.1** | 32.7 | 84.6 | 57.3 | 58.8 | 40.1 | 64.8 | 63.2 | 80.5 | 28.3 | 75.6 | 78.1 | 68.1 |
| U-Net | 55.7 | 36.9 | 77.4 | 68.6 | 20.7 | 20.0 | 41.7 | 22.2 | 42.3 | 62.1 | 50.5 | 51.1 | 43.2 |
| PANet | 48.2 | 48.4 | 78.2 | 30.4 | 31.9 | 53.2 | 53.8 | 42.1 | 32.6 | 66.2 | 72.2 | 67.8 | 46.7 |
| ESE | 26.6 | 50.7 | 57.3 | 21.7 | 47.9 | 19.0 | 41.7 | 62.3 | 37.4 | 55.2 | 37.1 | 25.2 | 38.6 |
| STS | 33.2 | 33.7 | 51.0 | 50.2 | 40.9 | 39.0 | 35.8 | 29.3 | 49.5 | 43.1 | 34.2 | 27.3 | 38.7 |

From Table 4, we can intuitively see that under the truck subclass, most algorithms achieve high accuracy, while under the person subclass, most algorithms have low accuracy. This is mainly because the person is a living body, and the amplitude of the movement of the living body generally changes greatly, plus the interference of the background. Therefore, the segmentation accuracy of person will be low.  Under the sub-categories such as bridge, bus, terrain, bicycle, and tunnel, the algorithm in this paper has achieved the best results. In terms of structure, DisRnet, SETR, U-Net, and Bisev2 all use context information fusion to achieve pixel-level segmentation, but the DisRnet network achieves the best results. Therefore, it can be concluded that the Dis pool and Rel pool have advantages. Then, most of the mentioned algorithms have higher accuracy than other algorithms, so it can also be concluded that the algorithm based on the context information fusion method has superiority in pixel-level semantic segmentation.

### 4.4.2 Pascal VOC

Compared with the Cityscapes dataset, Pascal VOC is a dataset with a more complex background and a larger amount of data, so the accuracy of some algorithms will be lower. Here we take the accuracy of the Cityscapes dataset as the benchmark, and subtract the accuracy of the PASCAL VOC dataset to obtain the difference in accuracy, as shown in Table 5 and Table 4 below. Through the analysis of Table 4 and Table 5, it is found that the accuracy of some algorithms is improved, and the accuracy of most algorithms is decreased. In the descending algorithm, the change in the accuracy of DisRnet is relatively small, and we can conclude that DisRnet has certain generalization ability. Through the analysis of Table 4 and Table 5, it can be seen that the accuracy of DisRnet has certain advantages compared with the compared algorithms, so it can be concluded that Dis pool and Rel pool have certain efficiency.

| Method | person | bird | cat | cow | dog | horse | sheep | boat | bus | chair | dining | table | mIou |
|---|---|---|---|---|---|---|---|---|---|---|---|---|---|
| DisRnet | 54.3 | 63.8 | **76.5** | 40.9 | 67.6 | 23.1 | 39.4 | 57.2 | 42.7 | 34.0 | **72.0** | **68.3** | **67.2** |
| SETR | 53.6 | 41.7 | 14.6 | 23.6 | 36.8 | 44.8 | 26.6 | **58.5** | 45.7 | 53.9 | 45.5 | 41.0 | 50.2 |
| GUDA | 59.1 | 46.7 | 42.1 | 53.2 | 39.8 | 29.1 | 35.9 | 52.0 | 13.6 | 39.0 | 18.3 | 46.7 | 54.6 |
| FedDG | 71.0 | 49.4 | 48.2 | 20.6 | 74.9 | 23.1 | 57.4 | 26.7 | 50.0 | 46.7 | 37.4 | 59.2 | 52.7 |
| DPL | 51.5 | 61.9 | 34.8 | **55.2** | **68.8** | 61.2 | 46.7 | 46.1 | 28.8 | 52.1 | 54.7 | 64.0 | 40.3 |
| CFPnet | **72.1** | **65.5** | 73.4 | 37.7 | 47.5 | 77.6 | 21.5 | 22.4 | **72.5** | **66.1** | 63.3 | 55.7 | 60.7 |
| Deep | 32.2 | 50.5 | 38.4 | 54.6 | 23.7 | 39.1 | **58.2** | 34.7 | 27.2 | 58.5 | 40.3 | 34.4 | 42.9 |
| Bisev2 | 51.2 | 32.6 | 51.8 | 50.0 | 45.4 | 61.8 | 47.3 | 31.5 | 46.8 | 59.6 | 53.5 | 60.7 | 60.1 |
| DFAnet | 48.1 | 25.1 | 40.2 | 27.3 | 50.3 | **62.4** | 21.2 | 25.3 | 27.8 | 44.9 | 35.8 | 48.9 | 55.2 |
| U-Net | 40.4 | 30.9 | 40.1 | 49.4 | 59.3 | 53.2 | 30.3 | 42.2 | 43.6 | 30.3 | 39.7 | 59.8 | 47.1 |
| PANet | 71.1 | 37.2 | 50.0 | 49.7 | 65.1 | 37.2 | 46.1 | 41.2 | 60.8 | 46.3 | 57.7 | 48.0 | 52.2 |
| STS | 38.0 | 50.2 | 38.3 | 43.2 | 51.6 | 48.7 | 30.5 | 47.2 | 71.9 | 65.3 | 34.7 | 28.5 | 48.1 |

Table 6. Difference between Cityscapes accuracy and Pacal VOC accuracy

| Method | DisRnet | SETR | GUDA | DPL | FedDG | CFPnet | Deep | Bisev2 | DFAnet |
|---|---|---|---|---|---|---|---|---|---|
| -(%) | 8.2 | 23.2 | 14.2 | 13.3 | -5.5 | 6.1 | -4.7 | 11.4 | 12.9 |

From Table 6, it can be seen intuitively that the difference of SETR has the largest change, 23.2%. Deep has the smallest difference change, 4.7%. However, combining Table 5 and Table 4, DisRnet has good generalization ability while ensuring high accuracy.

### 4.4.3 SBD

The SBD dataset is an enhanced dataset of PASCAL VOC. We use the model trained under PASCAL VOC as the training model, and then train and test it under the SBD dataset. The results are shown in Table 6, and the segmentation effect is shown in Figure 7.

Table 7. A comprehensive comparison of DisRnet and 5 SOTA algorithms under the SBD dataset.

| Network | DisRnet | SETR | FedDG | CFPnet | Bisev2 | DFAnet |
|---|---|---|---|---|---|---|
| Size(M) | 5.4 | 12.1 | 9.3 | 3.7 | 4.9 | 7.4 |
| mIoU(%) | 64.3 | 52.6 | 53.4 | 58.4 | 57.2 | 49.8 |
| GFLOPs | 20.3 | 26.8 | 17.6 | 14.7 | 18.6 | 13.4 |
| Fps | 78.4 | - | - | 26.2 | 89.2 | 59.2 |

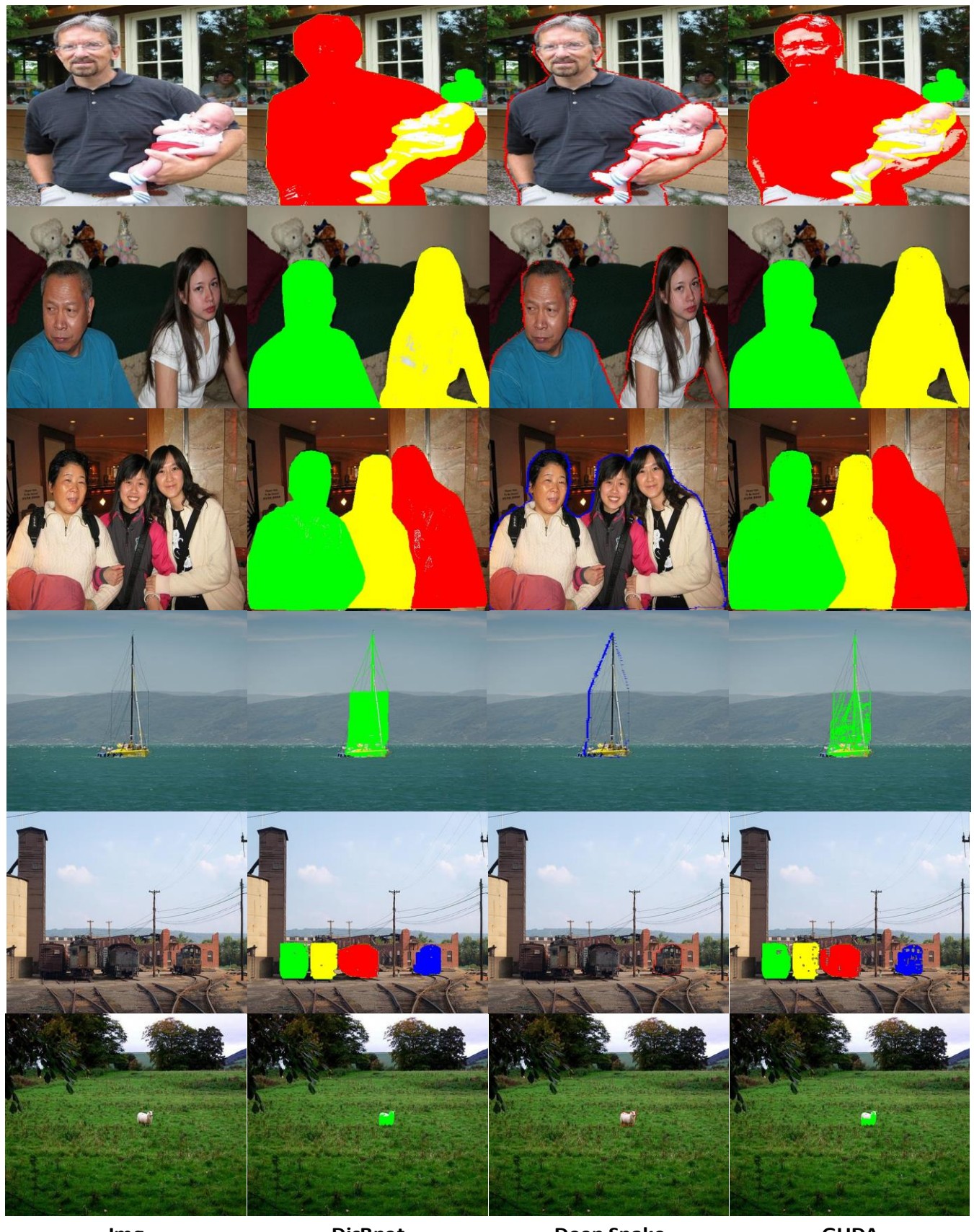

| Img | DisRnet | Deep Snake | GUDA |

Figure 7: Semantic effect segmentation effect of DisRnet, Deep Snake and GUDA under SBD dataset.

Under the Cityscapes dataset and the Pascal VOC dataset, we have done a comparative analysis of the accuracy. With the help of feature extraction of Dis pool and Rel pool and residual structure, DisRnet can achieve certain advantages. Now we conduct a full analysis under the SBD dataset. It can be seen from Table 7 that the GFLOPs of SETR are high, which mainly because SETR uses Transformers to obtain features, so the model of SETR is also larger. The models of CPFnet, Biesv2 and DisRnet are smaller. CPFnet uses two-channel pyramid pooling to extract features, which can obtain more features with less convolution, so the model is also smaller. And Bisev2 also extracts and fuses information through a dual-channel method. However, DisRnet adopts a residual structure in structure, which can effectively realize the lightweight of the model and improve the calculation speed at the same time.

Finally, in the segmentation comparison effect in Figure 7, we can see that when the background is too complex, DisRnet can also achieve pixel-level segmentation through related features. When the target is too small, DisRnet can obtain edge information about the target by analysing the correlation and discreteness between the target and the surrounding background, and finally achieve pixel-level segmentation.

## 5 CONCLUSION

In this paper, we design the Dis pool and the Rel pool by analysing the statistical characteristics of discrete data, and then combine the residual structure to design DisRnet. On the extracted features, the features extracted by the Dis pool and the Rel pool not only have more expressive, but also can reduce the loss of feature spatial information and improve the fault tolerance rate. The extracted features are fused by contextual information under the structure of residual, so as to achieve the effect of semantic segmentation. In ablation experiments, we demonstrate the feasibility of the Dis pool and the Rel pool. Then, we conduct a comprehensive comparison with some SOTA algorithms under the SBD, Cityscapes and Pascal VOC datasets，and find that DsiRnet performs pretty well in both accuracy and model size.

## 6 ACKNOWLEDGMENTS

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
