# OpenReview forum: "DisRnet: Discrete Relevance semantic segmentation network based on Discrete Data Distribution"
_graphicsinterface.org/Graphics_Interface/2022/Conference — Submitted to GI 2022_

### Official Review · Reviewer_2tGT · 2022-04-09
**DisRnet**

**Rating:** 2
**Confidence:** 4

**Review:**

The paper proposes a novel segmentation network architecture named DisRnet. The network contains 14 convolutional layers with some residual connection and self-attention links, overall the network has some similarity to ResNet-18. However, the authors design two kinds of new pooling layers. A Dis pooling operation which is to emphasize the detailed structure of the image and Rel pooling calculates the relation between windows in a feature map. The network also incorporates upsampling to ensure that the size of the segmentation map. Experiments are reported on Cityscapes, Pascal VOC and SBD. The authors compare DisRnet with good selection of SOTA algorithms on these datasets and show that the performance of DisRnet is comparable. They also argue for good generalization ability based on comparing Cityscapes and VOC results. Furthermore, they conduct an ablation where their replace their Dis and Rel pooling layers with max and average pooling.

Given the fairly small size and simple structure of the network, the results are quite competitive. As such, the pooling layers for self-attention may be of interest. Unfortunately, the description of the network is very poor and it can not be understood (some examples are below).


Page 1: "which eventually happen structural redundancy, large computational load, and segmentation errors in the semantic segmentation network"

"correlation between the data and the data"

"certain advantages in accuracy and speed."  which is never explained.

Page 2: "from the four indicators." which are never given.

Figure 1, there is no description on how these histogram for the real-valued feature maps after pooling were obtained.

"are smoother and more uniformly distributed. Therefore, the features extracted by the Dis pool and Rel pool are also easy to calculate."  Not sure how these observations are related.

Eqn. 2 misses a bracket.

Eqn. 3 definition of feature group missing.

Eqn. 4 where the largest integer smaller than a number is taken. Is this just the floor? Why is ceiling used?

Eqn. 6 seems to be correlation but subscripts and absolute value make no sense.

There is also no description what windows will be correlated. Neighboring windows, all windows etc.

Eqn. 7 is not motivated.

Table 1 and Figure 4 show no activation layers. Is this an oversight?

Table 1 does not have an explanation why dis and rel pooling have no channel number. Not clear why the final layer has 8 parameters.

Page 4 "let the input data a subtract the mean value of λ times, so that
the predicted value is smoother."  how would this result in a smoother map?

Page 4 "on the SBD training set for 20 epochs" Are 20 epochs enough for a fair comparison?

The claim "ResNet-18 represents the network structure after removing the pooling layer" is incorrect as ResNet-18 is a typical backbone network and does not contain upsampling.

The claim "Dis pool and the Rel pool is significantly better than the average layer and the max layer" is incorrect as no significance has been shown and worse Rel pooling is actually outperformed by average pooling.

Table 3 lambda = 0.7? Should this be 0.07?

Figure 6 uses u instead of lambda but worse has very different scale on the x-axis.

Page 5 "DisRnet varies with the value of a." Should this be lambda?

Page 6 "the accuracy of DisRnet has certain advantages compared with the compared algorithms,
so it can be concluded that Dis pool and Rel pool have certain efficiency." needs an explanation and justification.

Figure 7 shows no result for DeepSnake.

**References must be completely revised and properly formatted.**

[6] Technicolor T , Related S , Technicolor T , et al.  ...

[22] Hermann C , Spreitzer I , Schrder N W J , et al. Cytokine induction
by purified lipoteichoic acids from various bacterial species – Role
of LBP, sCD14, CD14 and failure to induce IL‐12 and subsequent
IFN ‐ γ release[J]. European Journal of Immunology, 2015,
32(2):541-551.
Not related to Linear Binary Patterns.

[55] Kaplan, D. , & Muñoz-Carpena, R. . (2011). Complementary effects
of surface water and groundwater on soil moisture dynamics in a
degraded coastal floodplain forest. 398(3), 221-234.

---

### Official Review · Reviewer_EFJH · 2022-04-13
**The paper presents a novel pixel-level image segmentation DNN. which extends ResNet with a so-called "discrete feature" layer and a correlation layer. Through experiments with some standardized data sets (Cityscapes, PASCAL VOC) they show the superiority of their approach.**

**Rating:** 3
**Confidence:** 2

**Review:**

The paper modifies an existing architecture for pixel-level segmenation of images with a "Dis pool" layer and a "Rel pool" layer. While the experimental validation looks promising, the details in the paper are hard to follow. Besides a number of grammar and spelling problems, some basics remain unclear that cloud a proper judgement of the ideas presented. E.g.

- Fig 1
  * How are the histograms computed: a) over what range, b) with what kernel? etc. Especially what is the "Dis" or the "Rel" kernel?
  * the y-scaling of the histograms differ, hence, any conclusion about the smoothness of these histograms cannot be verified since the scaling factor differs by an order of magnitude.

- Eq 1 is not a mean as claimed in the paper
- Eq 2 makes no sense, I assume we normalize by the mask-size N not by the running index "i*j"?
- Eq 5 makes no sense. l is a matrix, P is not well defined. Even if I should conclude from Fig 2 that P is a matrix of size HxW then those two matrixes don't have the same size. What operation is "*" here?
- The legend of Fig 6 talks about 'parameter u', but the text is referring to parameter lambda

--> because of these and other inaccuracies, I am not able to follow the paper and hence, I am not able to verify the logic behind the algorithm.

In addition, there are many inconsistencies in the references, in terms of details of the reference, capitalization, reporting of authors, etc.

Hence, at this point I cannot recommend the acceptance of the paper.

---

### Official Review · Reviewer_87Hi · 2022-04-13
**Presentation needs work**

**Rating:** 2
**Confidence:** 2

**Review:**

This paper proposes two pooling mechanisms to aggregate features in
convolutional networks for image segmentation. These pooling operators
are used in a residual network (ResNet) to replace maxpooling.

The premise and main contributions of the paper are not presented in a clear
and convincing manner. Although the tables indicate the proposed operators can
be beneficial, the evaluation does not speficy clearly the state of the art
baselines and the ablations, which makes it difficult to judge whether the
evaluation is meaningful at all. For example, "DisPool" and "RelPool" seem to
**not** downsample the feature maps, is the comparison to Average Pooling and
Max Pooling done under the same condition (i.e., de-activating their
downsampling)? It is not clear what is the training setup for the proposed
method (supervised for segmentation?). Are the baseliines retrained with the
same data and parameters?

In short, I think the paper is not ready for publication. Here a some high-level observations:

* The paper focuses on image segmentation. It seems better suited to a computer vision conference than Graphics Interface.
* I found the paper generally sloppy, very unclear and extremely hard to follow, if not unreadable. Even the abstract does not give a clear idea of
what exactly the paper proposes to solve. What does 'affected by the mean
pooling layer' mean? What exactly is the problem?
* The technical description is vague, unclear and difficult to follow. This makes reproduction difficult too.
* The writing needs work. The grammar is confusing at times, which makes it difficult to follow the paper.
E.g.,:
	- in the abstract: "these algorithms extract features, they are all affected by the mean pooling layer[...]".
	What does 'they' refer to? the algorithms or the features?
	- What are the discrete characteristics of discrete data distribution?
* It is not clear to me how the "Rel Pool" operator retains the spatial information
* I am surprised to see no citation to "Visualizing and understanding convolutional networks" [Zeiler 2013]
which introduces switch variables to preserve the spatial information in maxpooling layers, and unpooling layers
for reconstruction, or "Learning Deconvolution Network for Semantic Segmentation" [Noh 2015], which uses them
in a segmentation network.
* The premise that image data is discrete is questionable... The image is
quantized/discretized to be processed by a computer, using this aspect as the sole basis to
use discrete statistics is a bit of a stretch. It is not clear how the paper use the 'discrete' property at all.


### Some details to improvme the presentation

- Some equations (e.g. in 3.1) are stretched and pixelated in the rendered paper.
- U-Net fuses low/high resolution features, rather than low-dimensional... the dimension is typically understood
as the number of channels of the feature maps.
- Intro: 'which happens structural redundancy' -> what does that mean? why is it a problem?
- "the correlation coefficient of the feature by analyzing the correlation between the data and the data": what does that mean?
- Section2: "an image is composed of an indeterminate number of pixel values according to certain rules": what rules? This
is a rather limited definition of an image... (8bit in particular)
- Section 2: what are the limitations of HOG, SIFT, etc you refer to?
- The claim that CNN only do (simple) inner product is limited and misleading
- Figure1: histogram of what? feature values? color values?

Section 3.1:
- H, W and h, w are used interchangeably with different capitalizations
- What sliding window? it was not defined. What's the window size, stride, etc?
- Why does d_m need to be discrete?
- what is n? m?
- the Pi have first one index then 2? does that refer to deferent things? e.g. patch index vs. pixel indices within the patch?
- a_m does not look like an average, the denominator is missing
- what is d_m? why divide by i (or i*j)? there seems to be a parenthesizing issue in Eq. 2. for (i*j) or do you mean to divide by i and multiply by j?
- P_m(i,j) does not exist in Eq 1, 2
- what is the pixel value of Pm (singular). Isn't Pm a patch of pixels? Pm does
not shown in formula 3, what is the processing doing exactly? Can you explain
and motivate it?
- what is "the discrete value range"? how does this relate to \lambda*a_m + d_m?
- what is the high level motivation for this operator?
- I do not understand what you mean by Dis pool retains spatial information. Do
you mean there is no down/subsampling like in maxpool? Did you compare to using
maxpooling with overlapping windows (without downsampling)?. Figure 2 seems to suggest
there is no resampling happening.
- How does **not downsampling** improve the computational efficiency?
- Why is it a good thing to "reduce the value of feature map"? Do you mean the number of feature maps? their size?

Sectin 3.2:
- At this point I am very confused: what is the step size of the sliding window? Is is really reasonable to assume
that nearby pixels in a digital images are **independent** ?
- What is the relation between Pm and P_{m+1}? Do you always consider only the
- right neighbor patch? or do they refer to different **channels** in the
feature map? It would help to clarify the dimensions and domain of all the variables involved.
- Not clear why r "ensures the correlation".
- What is small "p" ?
- What does dot-multiplied mean? is it a point-wise product between tensors?

---

### Decision · Program_Chairs · 2022-04-17

Reject